# Diversity and antimicrobial activity of endophytic fungi isolated from *Securinega suffruticosa* in the Yellow River Delta

**Wen Du**[1,2,3,4]*, **Zhigang Yao**[1,3], **Jialiang Li**[3], **Chunlong Sun**[1,2,3,4]*, **Jiangbao Xia**[2], **Baogui Wang**[3], **Dongli Shi**[3], **Lili Ren**[3]

**1** Shandong Provincial Engineering and Technology Research Center for Wild Plant Resources Development and Application of Yellow River Delta, Binzhou University, Binzhou, China, **2** Shandong Key Laboratory of Eco-environmental Science for the Yellow River Delta, Binzhou University, Binzhou, China, **3** School of Biological and Environmental Engineering, Binzhou University, Binzhou, China, **4** Shandong Engineering and Technology Research Center for Fragile Ecological Belt of Yellow River Delta, Binzhou University, Binzhou, China

* duwen6688@163.com (WD); sunchunlong2016@163.com (CS)

**Data Availability Statement:** The data are all contained within the manuscript and/or Supporting Information files.

## Abstract

*Securinega suffruticosa* (Pall.) Rehd is an excellent natural secondary shrub in the Shell Islands of Yellow River Delta. The roots of *S. suffruticosa* have high medicinal value and are used to treat diseases, such as neurasthenia and infant malnutrition. Any organism that is isolated from this species is of immense interest due to its potential novel bioactive compounds. In this research, the distribution and diversity of culturable endophytic fungi in *S. suffruticosa* were studied, and the endophytic fungi with antimicrobial activity were screened. A total of 420 endophytic fungi isolates were obtained from the *S. suffruticosa* grown in Shell Islands, from which 20 genera and 35 species were identified through morphological and internal transcribed spacer (ITS) sequence analyses. *Chaetomium*, *Fusarium*, *Cladosporium*, and *Ceratobasidium* were the dominant genera. The high species richness S (42), Margalef index D′ (5.6289), Shannon–Wiener index H′ (3.1000), Simpson diversity index $D_s$ (0.9459), PIE index (0.8670), and evenness Pielou index J (0.8719) and a low dominant index λ (0.0541) indicated the high diversity of endophytic fungi in *S. suffruticosa*, the various species of endophytic fungi with obvious tissue specificity. The inhibition percentages of the 12 species of such endophytic fungi against *Colletotrichum siamense* were 3.6%-26.3%. *C. globosum*, *Fusarium* sp.3, and *C. ramotenellum* had a high antibacterial activity against *Enterococcus faecalis*, *Escherichia coli*, *Pseudomonas aeruginosa*, and *Staphylococcus aureus*. The Minimum Inhibitory Concentration (MIC) and the Minimum Bactericidal Concentration (MBC) were between 0.5 mg/mL and 2 mg/mL. Alkaloid content detection indicated that endophytic fungi had a high alkaloid content, whereas the alkaloid contents of *C. globosum* and *Fusarium* sp.3 reached 0.231% and 0.170%, respectively. Members belonging to the endophytic fungal community in the *S. suffruticosa* of Shell Islands that may be used as antagonists and antibacterial agents for future biotechnology applications were identified for the first time.

**Funding:** This study was supported by Shandong Provincial Natural Science Foundation ZR2019MH054, ZR2018PC010, ZR2019PH097, China; Doctor Foundation of Binzhou University 2016Y17, 2016Y02, China.

**Competing interests:** The authors have declared that no competing interests exist.

## Introduction

The Shell Islands of Yellow River Delta is one of the three largest old shell islands in the world; it is mainly formed by the shells and debris of shellfish living in the intertidal zone after their death, which are transported by waves and piled near the high tide line [1,2]. The beach ridge area of Shell Islands is relatively high above sea level, and the underground water level is low. Moreover, the shell sand soil has high porosity, the coarse sand content is the highest, and the retention precipitation capability is poor. Coupled with the high regional evaporation-precipitation ratio, the seasonal water shortage is serious, and the vegetation type is prioritized, specifically xerophytic shrubs and herbs. *Securinega suffruticosa* (Pall.) Rehd is a common shrub tree species in beach ridge zones [1,2].

*S. Suffruticosa* is a deciduous shrub belonging to the Euphorbiaceae family that has extremely strong adaptability and can endure cold, drought, and barren conditions, i.e., sandy soil [3]. *S. suffruticosa* grows approximately 1–2 m in height and has clusters, twigs, and tree-shaped expansion. Its roots are used in traditional Chinese medicine [4, 5]. It is rich in securinega-type alkaloids, especially securinine, and contains active compounds, such as flavonoids, rutin, tannins, phenolic compounds, and various amino acids [6]. Pharmacological studies have shown that *S. suffruticosa* can promote blood circulation, relax the muscles and tendons, invigorate the spleen and kidney, and relieve rheumatic pains. Moreover, this shrub is effective for the treatment of neurasthenia, facial nerve paralysis, post-polio syndrome, dizziness, deafness, narcolepsy, impotence, and acute liver injury, and other diseases [7–10]. It possesses an extremely high application value [4]. In addition to the plant itself, several studies on the microorganisms in the ecological environment of Shell Islands have reported the presence of actinomycete genera in the soil, specifically *Streptomyces* and *Nocardiopsis*. A total of 94 strains exhibit positive results in at least one antifungal or antibacterial assay, thereby suggesting that the Shell Islands of Yellow River Delta is a rich source of actinomycetes with many potential new species and active strains [11]. The fungi from the coastal saline soil in the Yellow River Delta grow in habitats under unique conditions due to the activation of metabolic pathways and the synthesis of distinct unknown molecules [12]. The production of these compounds supports the adaptation and survival of fungi in special ecosystems [13].

Endophytic fungi refer to a type of fungi inside plants that do not cause obvious plant diseases [14]. These fungi and their host plants have a very complex relationship. Some endophytic fungi can produce hormones that promote plant growth, such as anti-phagocytes that help the host resist biological feeding, develop medicinal ingredients, and produce many products with biological activities [15]. Endophytic fungi with antimicrobial activity of natural products can compensate for the lack of plant resources, the regeneration cycle length limit, and the use of industrial fermentation to produce natural active compounds for mass production at low cost and no pollution [16]. Many scholars have discovered novel structures and the antimicrobial activities of metabolites from endophytic fungi. A new family of 4-hydroxy-2-pyridone has been isolated from a mangrove endophytic fungus *Campylocarpon* sp. HDN13-307 [17]. Vincamine, a monoterpenoid indole alkaloid that significantly inhibits acetylcholinesterase activity was isolated from an oleander endophytic fungus *Geomyces* sp. CH1 [18]. Endophytic fungi are widely distributed in many plant species across all plant groups and play an important role in bio-antagonism and abiotic stresses. Furthermore, plant tolerance of biological stress is related to the natural products of these endophytic fungi. Many strains can produce various active natural products [19]. Approximately half of the newly discovered fungal natural products are derived from endophytic fungi, including numerous natural compounds with antimicrobial activity [20]. Endophytic fungi are new sources of novel active compounds with biological activity and are subjected to biotechnological developments,

but their true potential remains underexplored [21]. Moreover, knowledge about the endophytic fungal community in *S. suffruticosa* is insufficient. Therefore, this study aimed to determine the species diversity of the endophytic fungal community in *S. suffruticosa*, a common shrub species in the Shell Islands of Yellow River Delta, China. Results may lead to the discovery of new species or strains with valuable bioactive compounds. In this research, the diversity of the endophytic fungi isolated from different tissues of healthy *S. suffruticosa* was evaluated, and their potential antimicrobial activities against various pathogens were examined. To the best of our knowledge, this work is the first report on the biodiversity, phylogeny, and assessment of the antimicrobial activity of the endophytic fungi harbored in *S. suffruticosa*.

## Materials and methods

### Chemicals, microorganism, media, and culture conditions

In April 2017 and May 2018, healthy *S. suffruticosa* plants of 3–6 years were selected in Shell Islands of the Yellow River Delta, and authenticated by Professor Jingkuan Sun from Binzhou University. The collections of plants were approved by Binzhou Forestry Bureau. *Colletotrichum siamense*, *Staphylococcus aureus*, *Escherichia coli*, *Pseudomonas aeruginosa*, *Enterococcus faecalis*, *Fusarium oxysporum*, *Phoma herbarum* and *Colletotrichum siamense* were used as the test pathogen. The test strains were provided by the Biopharmaceutical Center of Binzhou University and the institute of Biochemistry and Nutrition of Guizhou University. DNA extraction kit, 2×Pfu PCR premix solution were purchased from Chengdu Rambo Biotechnology Co., Ltd. DNA markers, primers ITS1 (5′–TCCGTAGGTGAACCTGCGG–3′) and ITS4 (5′–TCCTCCGCTTATTGA TATGC–3′) were synthesized by Biotech Bioengineering (Shanghai) Co. Ltd. [22]. Securinine was purchased from Sichuan Weikeqi Biological Technology Co., Ltd., Other reagents used were analytically pure.

### Separation and purification of culturable endophytic fungi

The endophytic fungi were separated and purified following the methods reported [23]. The samples were washed repeatedly under running tap water, removing the surface soil and appendages. 3–6 years of fresh and healthy *S. suffruticosa* were bigger, intercepted of some tissue for cleaning. The more robust main roots, stems and leaves were selected, and then rinsed off after a small section of about 10 cm. The roots, stems, and leaves of *S. suffruticosa* were cut into small pieces of about 0.5 cm and placed in clean petri dishes. Surface disinfection was carried out according to the following procedures: the samples were rinsed with 75% ethanol for 2–3 min and then with sterile water for 4–6 times. They were disinfected with 0.1% mercuric chloride for 3–5 min and rinsed with sterile water for four to six times. Excess water was soaked with filter paper, and the material was cut into 0.5 cm × 0.5 cm small pieces (slices) using sterile technique. The abovementioned tissue blocks were then placed in PDA medium supplemented with 100 U/mL penicillin with three blocks in each petri dish and incubated at a constant temperature of 25°C. At the same time, the above-mentioned surface-sterilized materials were directly planted in PDA medium and incubated at 25°C to check whether the surface was completely disinfected. After the cultivation of 3–14 days, the fungal hyphae on the edge of the tissue block was picked out in time and transplanted to fresh PDA medium for cultivation and purification. After purification, the colonies were transferred to the agar slant culture medium for future use.

### Molecular recognition of endophytic fungi

The endophytic fungi were taken after fermentation and centrifuged. A total of 100 mg hyphae was taken and placed a 2 mL microcentrifuge tube. The DNA of endophytic fungi was

extracted using CTAB method according to instructions of the DNA extraction kit (Biotech Bioengineering Co. Ltd.). Based on the 25 μL system, 12.5 μL of 2 × Pfu PCR Mix, 0.5 μL of ITS4, 0.5 μL of ITS5, 1.5 μL of DNA, and 10 μL of ultrapure water were added. The ITS1-5.8S rDNA-ITS2 sequence was amplified at 95˚C for 5 min, 95˚C for 45 s, 52˚C for 30 s, and 72˚C for 3 min, for 35 cycles, then at 72˚C for 15 min [22]. The PCR reaction product was subjected to electrophoresis at 120 V for 30 min using 1% gel, and the presence of 400–800 bp fragment was determined by markers. Samples of fragments after successful amplification were sent to Biotech Bioengineering (Shanghai) Co., Ltd., for purification, and sequencing. According to the sequencing results, BLAST alignment was performed in the GenBank database to search for homologous sequences to avoid misidentification [24]. Representative isolates were selected for analysis. All ITS sequences were submitted to the GenBank database under accession number MH383162- MH383218.

## Data analysis of the endophytic fungi

The relative frequency (RF) and dominance were used to describe quantitatively the abundance, distribution preference, and composition of the endophytic fungi taxa in *S. suffruticosa*. The relative frequency (RF) was the ratio of the number of isolates of a certain genus or taxon to the total number of isolates, according to the Eq (1). Dominance is calculated using the Eq (2), when dominance Y > 0.02, the genus is the dominant genus [25, 26]. Species as a statistical unit was used to calculate the number of isolates (N). Species richness was evaluated by the species richness index (S) and Margalef index (D′), two important parameters of diversity analysis [27]. Species richness index (S) was obtained by counting the number of endophytic fungal species in each part of the plant. Species richness index (S) was obtained by calculating the number of endophytic fungi per tissue or total plant. The Margalef index (D′) was calculated according to Eq (3). Species diversity was evaluated by Shannon-Wiener index (H′), Simpson diversity index ($D_S$) and Simpson′s dominant index (λ) [27]. Shannon-Wiener index (H′), Simpson′s diversity index (Ds) and Simpson′s dominant index (λ) were respectively calculated by Eqs (4–6). The probability of interspecies encounter (PIE) index was used to assess the probability of individuals belonging to different species [28]. PIE index was calculated by Eq [7]. Species evenness was evaluated by Pielou′s evenness index (J) [29], which was calculated by Eq (8). By Eq (9) calculated the relative abundance (RA) of each of the genus:

$$RF(\%) = (n_i/N_t) \times 100\% \tag{1}$$

$$Y = (n_i/N_t) \times f_i \tag{2}$$

$$D' = (S-1)/\ln N_t \tag{3}$$

$$H' = -\sum_{i=1}^{s} P_i \ln P_i, \quad P_i = N_i/N_t \tag{4}$$

$$D_s = 1 - \sum_{i=1}^{s} P_i^2 \tag{5}$$

$$\lambda = \sum_{i=1}^{s} p_i^2 \tag{6}$$

$$PIE = \sum_{i=1}^{s}(N_i/N_t)(N_t - N_i)/(N_t - 1) \tag{7}$$

$$J = H/H_{max}, \quad H_{max} = \ln S \tag{8}$$

$$RA(\%) = N'/N_t \times 100\% \tag{9}$$

where $N_t$ represents the total number of isolates of all genera obtained by separation, $N_i$ is the number of isolates belonging to the i-th species, $n_i$ is the number of isolates of the i-th genus, and $f_i$ is the frequency of occurrence of this genus in different *S. suffruticosa* plants. S is the total number species in each tissue or total plant, $N'$ is the number endophytic fungal isolates from each genera.

## Antifungal non-volatile compounds test

The effects of non-volatile metabolites produced by the selected endophytic fungi were determined using the method described by Hajieghrari et al. [30]. To select endophytic fungi hyphae or spores, according to the amount of 10% (V/V), were inoculated in PDA medium, and cultured under 25˚C on an orbital *shaker* at 120 r/min for seven days. They were then centrifuged at 5000 r/min for 20 min to obtain culture broth. The filtrate was poured into the Petri dish at a final concentration of 20% (v/v). Once solidified, a 5mm disc of the test pathogen was placed in the center of the PDA plate; they were cultured at 25 ˚C. The control plates were prepared by culture filtrate without amending PDA. After 7 days of observation, the mycelial radial growth of the tested pathogen on a control plate (r1) and the direction of antagonistic fungi (r2) and measured and inhibited percentage (%) in mycelial growth was calculated according to the formula (30): *I*% = [(r1—r2)/r1] x 100. All experiments were performed using at least three replicates. The data presented correspond to mean values, the standard deviation being lower than 15%.

## Antibacterial assay

Filter paper dispersion method was used: bacteria indicator solution (about $1 \times 10^8$ cfu/mL) was applied to the beef extract peptone plate, and the sterilized filter paper was applied to an appropriate place. Subsequently, 20 μL of the fermentation broth was absorbed and placed on the filter paper. Sterile water was used as a control group, the process was repeated three times for each sample. The bacteria were cultured at 37˚C for 16–24 h. The cell growth and size of the inhibition zone were observed [31, 32]. All experiments were performed using at least three replicates. The data presented correspond to mean values, the standard deviation being lower than 15%.

## Determination of minimum inhibitory concentration (MIC) and minimum bactericidal concentration (MBC)

Ethyl acetate extract with endophytic fungi of 10 g dry mycelium twice, the extract was concentrated under 40˚C decompression using a rotary vacuum evaporator. MIC and MBC were determined by modified microdilution two-fold assay using 96-well assay plate. Three 96-well enzyme plates were taken and soaked overnight in a beaker containing 70% ethanol before use. Take it out the next day, dry it carefully in the drying box, and put it in the ultra-clean workbench. In each test hole, beef extract peptone liquid medium, suspension of bacteria and

crude extract of endophytic fungi were added to mix well. Then, appropriate amount of the first hole was added into the second hole, and so on. Bacteria suspension was not added in the control group.

Seal after mixing, placed in the whole of the 37 ˚C temperature oscillation incubator for the night. MIC of endophytic fungi was recorded on day 2. Sterile medical cotton swab was dipped into the liquid in the enzyme label plate in the ultra-clean workbench, and the liquid was uniformly coated on the beef extract peptone medium. Overnight, the second day was taken out to observe the growth of bacteria and record the MBC of endophytic fungi [33].

## Measurement of total alkaloid content

The total alkaloid content was measured following the methods reported [25]. Approximately 2 g of dry mycelia was placed in a conical flask with a cover and then mixed with 2 mL 18% ammonia liquor. After 1 h, a 30 mL mixture containing ethyl ether, chloroform, and ethanol (25:8:2.5) was added. Ultrasonic extraction was performed for 20 min, and 30 mL of the obtained supernatant was cold-soaked for 30 min before being subjected to ultrasonic concussion and 20 min extraction. The product was filtered and rinsed with 15 mL of the same solvent thrice before being mixed with the filtrate. The resulting mixture was placed in a 60˚C water bath until it dried. Chloroform (10 mL) was added, and 5 mL of the product was transferred to a small separating funnel. Then, 6 mL chloroform and 2 mL buffer solution (25 mL of 0.2mol/L potassium hydrogen phthalate and 11.83 mL of 0.2 mol/L sodium hydroxide; pH = 5.0) were added. A 0.001 mol/L bromothymol blue solution was used for the titration, and the mixture was shaken constantly. Finally, 5 mL fresh chloroform was added after the separation of the chloroform layer. The mixture was titrated, shaken, and then allowed to set until the water layer showed a slightly yellowish color. The total alkaloid content was calculated according to the following formula: $Y\% = (NVM \times 2)/W \times 100\%$, where $N$ denotes the concentration of the bromothymol blue solution (mol/L), $V$ refers to the volume of consumed bromothymol blue solution (L), $M$ is the molecular weight of securinine (217.26), and $W$ represents the sampling weight (g).

## Results

### Identification of culturable endophytic fungi from *S. suffruticosa*

A total of 420 endophytic fungal isolates were obtained from *S. suffruticosa*. Specifically, 143 isolates originated from stems, 170 from roots and 107 from leaves. The number of endophytic fungi in the roots of *S. suffruticosa* was significantly higher than those in the leaves, indicating that the distribution of endophytic fungi differs among the different plant tissues. Many isolates had the same morphological characteristics. A total of 57 morphotypes were selected for molecular identification. DNA extraction, PCR amplification, target bands purification and sequencing were performed on the endophytic fungal isolates from *S. suffruticosa*. All sequences were BLAST aligned and submitted to NCBI data (S1 Table). For identification and clustering, we compared the sequences of 57 endophytic fungi with the nearest species with the maximum-likelihood (ML) method. As shown in Fig 1, the endophytic fungal phylogenetic clustering is consistent with their identification at the species level. Although the isolates G3 (G7, G11, G13, G14), G2(G10), G1(G9), and G5 were clustered on the same branch, the support rating was only 61%, thereby indicating that they were also different. The isolates Y1, Y4, Y6, Y7, and Y8 and *Fusarium solani* were clustered on the same branch, with 68% support rating. Thus, they were considered different species. The isolates Y2(Y5) and Y3 were clustered on different branches and subsequently divided into two different species. The isolates C1, C2, and C3 and many strains of *Colletotrichum* sp. were clustered on the same branch. Hence, they

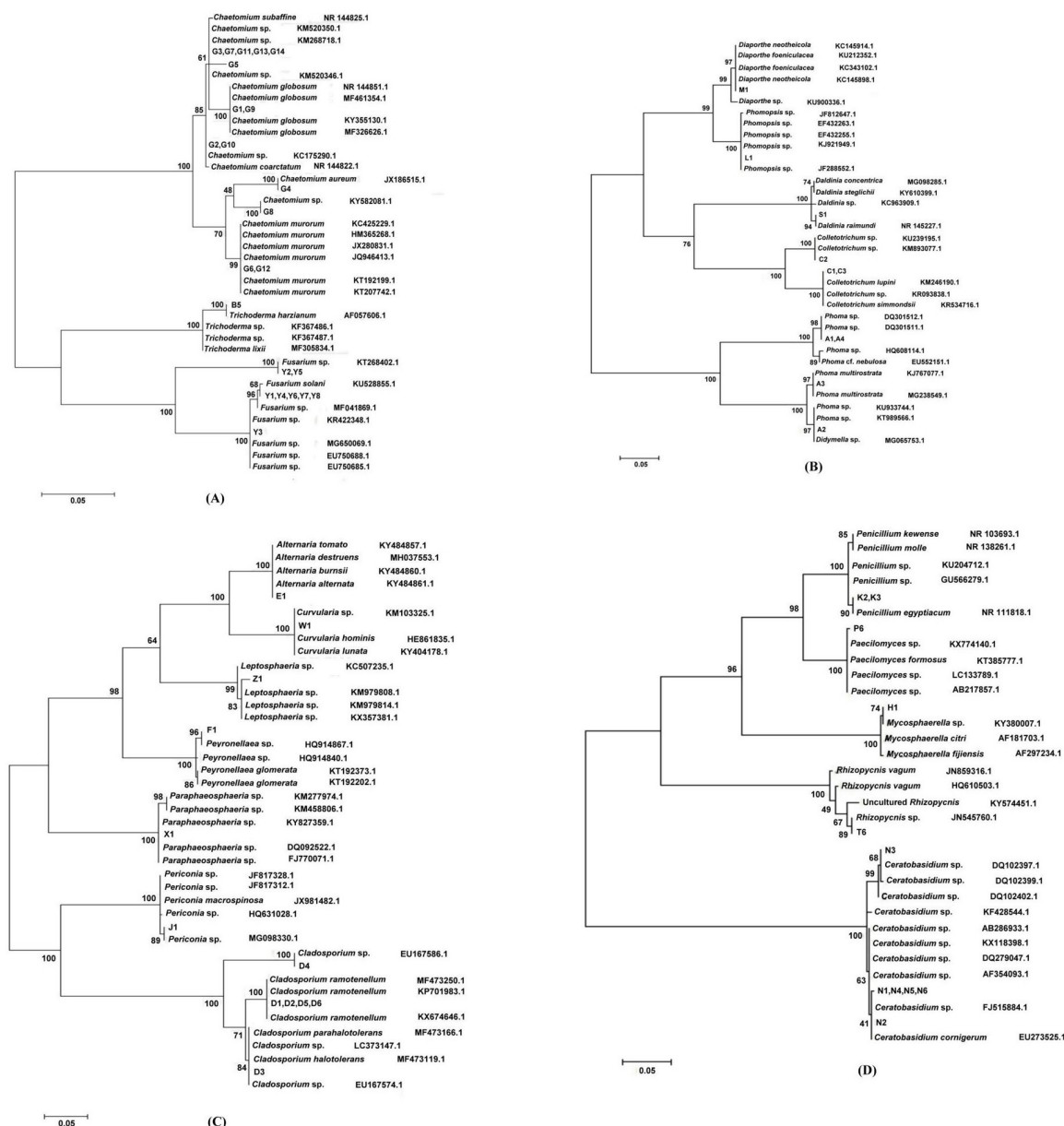

**Fig 1. Maximum-likelihood phylogenic analyses by internal transcribed spacer (ITS) sequence alignment for the endophytic fungi from *S. suffruticosa* belonging to Sordariales and Hypocreales (A); Phyllachorales, Diaporthales and Xylariales (B); Capnodiales and Pleosporales (C); Mycosphaerellales, Incertae sedis *Rhizopycnis*, Eurotiales and Cantharellales (D).** The tree has been drawn to scale (0.05).

all belonged to *Colletotrichum*. Isolates C1(C3) and C2 belonged together on different branches. Hence, they were divided into two different species. The isolate A3 and *Phoma multirostrata*, with 97% of the support ratings on the same branch, identified them as belonging to the same species. The isolates N1, N4, N5, N6, N2, and N3 and many strains of *Ceratobasidium* sp. were in the same branch. Hence, they belonged to *Ceratobasidium*. Although N1(N4, N5, N6) and N2 are clustered on the same branch, the support rating was only 41%. Hence, N3, N1, N4, N5, N6, and N2 are divided into three different species. In addition, *Didymella* in the anamorphic stage was called *Phoma* [27]. The teleomorphic and anamorphic fungi with

different generic names have the same genetics. However, we did not combine the morphological characteristics of teleomorphic and anamorphic fungi.

Based on the rDNA-ITS sequence analysis, 420 isolates of endophytic fungi were identified into two phyla, four classes, 11 orders, and 20 genera (Table 1). All endophytic fungi were distributed into Ascomycota and Basidiomycota. Basidiomycota only includes the *Ceratobasidium* in the order Cantharellales class Agaricomycetes. The other 19 genera of endophytic fungi belong to three classes in Ascomycota as follows: class Sordariomycetes, including

**Table 1. Composition of endophytic fungi from roots, stems and leaves of S. *suffruticosa*.**

| Phylum | Class | Order | Genus | Numbers of isolates (relative frequency) | Species | Morphotype | Numbers of isolates | | |
|---|---|---|---|---|---|---|---|---|---|
| | | | | | | | Root | Stem | Leave |
| Ascomycota | Sordariomycetes | Sordariales | *Chaetomium* | 120(28.57%) | *Chaetomium globosum* | G1,G9, | 21 | 1 | 6 |
| | | | | | *Chaetomium* sp. 1 | G2,G10 | 5 | 5 | 5 |
| | | | | | *Chaetomium* sp. 2 | G3,G7,G11,G13,G14 | 27 | 3 | 4 |
| | | | | | *Chaetomium aureum* | G4 | 2 | 5 | 1 |
| | | | | | *Chaetomium* sp. 3 | G5 | 6 | 7 | 2 |
| | | | | | *Chaetomium murorum* | G6,G12 | 8 | 6 | 0 |
| | | | | | *Chaetomium* sp. 4 | G8 | 3 | 1 | 2 |
| | | Hypocreales | *Fusarium* | 82(19.52%) | *Fusarium* sp. 1 | Y2,Y5 | 18 | 11 | 0 |
| | | | | | *Fusarium* sp. 2 | Y3 | 3 | 4 | 0 |
| | | | | | *Fusarium* sp. 3 | Y1,Y4,Y6,Y7,Y8 | 27 | 19 | 0 |
| | | | *Trichoderma* | 1(0.24%) | *Trichoderma harzianum* | B5 | 1 | 0 | 0 |
| | | Phyllachorales | *Colletotrichum* | 23(5.48%) | *Colletotrichum* sp. 1 | C1,C3 | 0 | 12 | 6 |
| | | | | | *Colletotrichum* sp. 2 | C2 | 0 | 2 | 3 |
| | | Diaporthales | *Diaporthe* | 2(0.48%) | *Diaporthe* sp. | M1 | 0 | 2 | 0 |
| | | | *Phoma* | 38(9.05%) | *Phoma* sp. 1 | A1,A4 | 0 | 10 | 6 |
| | | | | | *Phoma* sp. 2 | A2 | 0 | 2 | 4 |
| | | | | | *Phoma multirostrata* | A3 | 1 | 8 | 7 |
| | | | *Phomopsis* | 8(1.90%) | *Phomopsis* sp. | L1 | 0 | 3 | 5 |
| | | Xylariales | *Daldinia* | 3(0.71%) | *Daldinia* sp. | S1 | 0 | 2 | 1 |
| | Dothideomycetes | Capnodiales | *Cladosporium* | 66(15.71%) | *Cladosporium ramotenellum* | D1,D2,D5,D6 | 10 | 13 | 31 |
| | | | | | *Cladosporium* sp. 1 | D3 | 2 | 0 | 3 |
| | | | | | *Cladosporium* sp. 2 | D4 | 1 | 3 | 3 |
| | | Pleosporales | *Alternaria* | 6(3.37%) | *Alternaria* sp. | E1 | 0 | 3 | 3 |
| | | | *Peyronellaea* | 1(0.24%) | *Peyronellaea* sp. | F1 | 1 | 0 | 0 |
| | | | *Periconia* | 3(0.71%) | *Periconia* sp. | J1 | 2 | 1 | 0 |
| | | | *Curvularia* | 7(1.67%) | *Curvularia* sp. | W1 | 2 | 2 | 3 |
| | | | *Paraphaeosphaeria* | 6(1.43%) | *Paraphaeosphaeria* sp. | X1 | 2 | 1 | 3 |
| | | | *Leptosphaeria* | 5(1.19%) | *Leptosphaeria* sp. | Z1 | 2 | 2 | 1 |
| | | Mycosphaerellales | *Mycosphaerella* | 1(0.24%) | *Mycosphaerella* sp. | H1 | 1 | 0 | 0 |
| | | Incertae sedis | *Rhizopycnis* | 3(0.71%) | *Rhizopycnis* sp. | T6 | 3 | 0 | 0 |
| | Eurotiomycetes | Eurotiales | *Paecilomyces* | 1(0.24%) | *Paecilomyces* sp. | P6 | 1 | 0 | 0 |
| | | | *Penicillium* | 8(1.90%) | *Penicillium* sp. | K2,K3 | 3 | 2 | 3 |
| Basidiomycota | Agaricomycetes | Cantharellales | *Ceratobasidium* | 36(8.57%) | *Ceratobasidium* sp. 1 | N1,N4,N5,N6 | 14 | 11 | 3 |
| | | | | | *Ceratobasidium* sp. 2 | N2 | 2 | 1 | 1 |
| | | | | | *Ceratobasidium* sp. 3 | N3 | 2 | 1 | 1 |

*Chaetomium* in the order Sordariales, *Fusarium* and *Trichoderma* in the order Hypocreales, *Colletotrichum* in the order Phyllachorales, *Diaporthe*, *Phoma* and *Phomopsis* in the order Diaporthales, and *Daldinia* in the order Xylariales; class Dothideomycetes, including *Cladosporium* in the order Capnodiales, *Alternaria*, *Periconia*, *Curvularia*, *Paraphaeosphaeria*, *Neosetophoma*, and *Peyronellaea* in the order Pleosporales, *Mycosphaerella* in the order Mycosphaerellales, and *Rhizopycnis* in an undecided order; and class Eurotiomycetes, including *Paecilomyces* and *Penicillium* in the order Eurotiales. Thirty-six *Ceratobasidium* isolates can be divided into three species, 120 *Chaetomium* isolates into seven species, 82 *Fusarium* isolates into three species, 38 *Phoma* isolates into three species, 66 *Cladosporium* isolates into three species, and eight *Penicillium* isolates into two species. In addition, three isolates belonged to the species of *Rhizopycnis* of an unknown family (Table 1).

## Diversity analyses of culturable endophytic fungi from *S. suffruticosa*

The endophytic fungi of *S. suffruticosa* were composed of fungi in Dothideomycetes and Sordariomycetes. The Pleosporales fungi in Sordariales and Hypocreales of Dothideomycetes were the most common endophytic fungi in *S. suffruticosa*, with a relative frequency of 48.33%. The relative frequencies of *Chaetomium* in Sordariales, *Fusarium* in Hypocreales, and *Cladosporium* in Capnodiales and *Ceratobasidium* in Cantharellales were 28.57%, 19.52%, 15.71%, and 8.57%, and their dominance values Y were 0.2300, 0.1367, 0.0707, and 0.0300, respectively. The species composition and Y value of endophytic fungi (Tables 1 and 2) in the root, stem, and leaf tissues of *S. suffruticosa* indicated that at least 15 genera and 33 species of endophytic fungi were present in the root tissues. The dominant species were *Chaetomium*, *Fusarium*, and *Ceratobasidium*, and their Y values were 0.3388, 0.1129 and 0.0371, respectively. In the stem tissue, at least 13 genera and 33 species of endophytic fungi were present and dominated by *Fusarium*, *Chaetomium* and *Cladosporium* with Y values of 0.1664, 0.1077,

**Table 2. The dominance (*Y*) values of endophytic fungi from roots, stems and leaves of *S. suffruticosa*.**

| Genus | Roots | Stems | Leaves | Total |
|---|---|---|---|---|
| *Chaetomium* | 0.3388 | 0.1077 | 0.0654 | 0.2300 |
| *Fusarium* | 0.1129 | 0.1664 | / | 0.1367 |
| *Trichoderma* | 0.0003 | / | / | 0.0001 |
| *Colletotrichum* | / | 0.0196 | 0.0168 | 0.0164 |
| *Diaporthe* | / | 0.0007 | / | 0.0002 |
| *Phoma* | 0.0003 | 0.0140 | 0.0318 | 0.0181 |
| *Phomopsis* | / | 0.0021 | 0.0070 | 0.0029 |
| *Daldinia* | / | 0.0007 | 0.0005 | 0.0004 |
| *Cladosporium* | 0.0153 | 0.0336 | 0.1556 | 0.0707 |
| *Alternaria* | / | 0.0021 | 0.0014 | 0.0014 |
| *Peyronellaea* | 0.0003 | / | / | 0.0001 |
| *Periconia* | 0.0006 | 0.0003 | / | 0.0004 |
| *Curvularia* | 0.0006 | 0.0007 | 0.0028 | 0.0017 |
| *Paraphaeosphaeria* | 0.0006 | 0.0003 | 0.0028 | 0.0014 |
| *Leptosphaeria* | 0.0006 | 0.0007 | 0.0005 | 0.0006 |
| *Mycosphaerella* | 0.0003 | / | / | 0.0001 |
| *Rhizopycnis* | 0.0009 | / | / | 0.0004 |
| *Paecilomyces* | 0.0003 | / | / | 0.0001 |
| *Penicillium* | 0.0018 | 0.0007 | 0.0028 | 0.0019 |
| *Ceratobasidium* | 0.0371 | 0.0182 | 0.0093 | 0.0300 |

and 0.0336. At least 12 genera and 29 species of endophytic fungi were found in the leaf tissue and dominated by *Cladosporium*, *Chaetomium*, and *Phoma* with Y values of 0.1556, 0.0654, and 0.0318.

Table 3 summarizes the indices related to the diversity of endophytic fungi in *S. suffruticosa*. The species richness (S) and Margalef index (D′) can reflect the richness of endophytic fungal species [34]. High values of S and D′ indicate abundant endophytic fungal species. In this experiment, species diversity was analyzed based on Shannon–Wiener index (H′), Simpson diversity index ($D_s$), and probability of interspecific encounter index (*PIE*). These indices showed the heterogeneity/homogeneity of the frequency of the emergence of species. In general, a high H′ corresponds to $D_s$ close to 1. In terms of diversity index, the *H′* and $D_s$ of endophytic fungi in *S. suffruticosa* were 3.1000 and 0.9459, respectively, and *PIE* was 0.8670. However, the diversity of endophytic fungi varied in different tissues of *S. suffruticosa*. In general, the *H′* and $D_s$ indices exhibited the same changes in different tissues and were the highest in the stem, followed by the leaves and roots due to the isolate number, isolation frequency, and species richness. The PIE value was related to *species* number *and individual* number of endophytic fungi and was significantly related to the uniformity distribution of plant species and the number of samples [35]. High *PIE* values showed that the habitat characteristics of different parts did not limit the growth of endophytic fungi [35]. The Pielou index (J), H′, and the size of sample reflected species uniformity. In this study, endophytic fungi from the stems showed the highest Pielou index and species richness. The dominant index (λ) was used to assess the ecological dominance of a community and was negatively correlated with $D_s$. A high λ value obtained in one part indicates that the endophytic fungal community may have less diversity and balance of species. The endophytic fungal community in the stem showed the lowest degree of ecological dominance with a λ value of 0.0661. In general, endophytic fungal communities from different isolated parts indicate the structure, richness, diversity, and dominance of different species. All endophytic fungi were isolated from three different tissues and thus could represent the entire endophytic fungal community. As shown in Table 3, the total endophytic fungi associated with *S. suffruticosa* exhibited high richness and diversity of species but low ecological dominance, with high S (35), D′ (5.6289), H′ (3.1000), $D_s$ (0.9459), and PIE (0.8670) and low λ (0.0541).

## Antagonistic activity of the fungal isolates against pathogenic fungi

Thirty-five species were selected according to the results of ITS sequence analysis. Dual culture test showed the inhibition percentage (I%) of various endophytic fungi on plant pathogenic fungi (Table 4). The antagonistic ability of endophytic fungi was determined by producing non-volatile compounds to inhibit the test pathogen. In plates containing endophytic fungal

**Table 3.  Diversity analyses of endophytic fungi.**

| Diversity Index | Different Tissues | | | Total |
|---|---|---|---|---|
| | Root | Stem | Leaf | |
| Species richness (S) | 27 | 29 | 24 | 35 |
| Margalef index (D′) | 5.0625 | 5.6419 | 4.9221 | 5.6289 |
| Shannon–Wiener index (H′) | 2.3369 | 3.0019 | 2.7228 | 3.1000 |
| Simpson diversity index ($D_s$) | 0.8774 | 0.9339 | 0.8878 | 0.9459 |
| PIE index (PIE) | 0.9113 | 0.8701 | 0.8961 | 0.8670 |
| Dominant index (λ) | 0.1226 | 0.0661 | 0.1122 | 0.0541 |
| Pielou index (J) | 0.7091 | 0.8915 | 0.8567 | 0.8719 |

**Table 4. In vitro antagonism of endophytic fungal isolates against the pathogenic fungi.**

| Isolate name | Percent inhibition growth over control (I%) | | |
|---|---|---|---|
| | *Fusarium oxysporum* | *Phoma herbarum* | *Colletotrichum siamense* |
| *Chaetomium globosum* | 16.1±1.2 | 21.4±1.3 | 26.3±1.6 |
| *Chaetomium* sp. 2 | - | - | 17.5±0.7 |
| *Chaetomium murorum* | - | 5.2±0.6 | 10.2±0.4 |
| *Fusarium* sp. 3 | 17.1±1.8 | 16.0±1.4 | 23.3±2.7 |
| *Fusarium* sp. 1 | - | - | 19.4±1.2 |
| *Fusarium* sp. 2 | 9.8±1.1 | 14.6±1.3 | 4.5±0.7 |
| *Phomopsis* sp. | - | - | 8.3±0.5 |
| *Cladosporium ramotenellum* | 18.9±0.6 | 14.6±0.8 | 21.6±0.6 |
| *Periconia* sp. | - | - | 3.6±0.5 |
| *Curvularia* sp. | - | 3.2±0.4 | 5.8±0.4 |
| *Leptosphaeria* sp. | - | - | 18.6±0.7 |
| *Penicillium* sp. | - | - | 17.4±1.3 |

filtrates, the radial growth of *C. siamense* mycelia showed significant I% for *C. globosum* (26.3%), *Fusarium* sp. 3 (23.3%), and *C. ramotenellum* (21.6%). The radial growth of *P. herbarum* mycelia showed significant I% for *C. globosum* (21.4%). The radial growth of *F. oxysporum* mycelia showed significant I% for *C. ramotenellum* (18.9%) (Table 4).

## Antibacterial properties

Among 35 species of endophytic fungi, nine (accounting for 25.71%) had significant inhibitory activity against *E. coli*. Four, six, and six species of fungi had significant inhibitory activity on *S. aureus*, *P. aeruginosa* and *E. faecalis*, respectively. Endophytic fungi *C. globosum*, *Fusarium* sp. 3, and *C. ramotenellum* had relatively good bacteriostatic effect (Table 5). Many endophytic fungi showed a significant inhibitory effect on various test pathogens, thereby exhibiting significant antimicrobial extensibility. The ethyl acetate extracts of the fungi were used to determine the minimum concentration required to inhibit bacterial growths. The crude extracts of *C. globosum* and *Fusarium* sp. 3 (1.00 mg/mL) were enough to inhibit *S. aureus*. The inhibition of *E. coli*, *P. aeruginosa* and *E. faecalis* required a crude extract concentration of at least 0.50 mg/mL (Table 6). The ethyl acetate extracts of *C. globosum* showed the best bacteriostatic effect.

**Table 5. Antibacterial activity of endophytic fungal isolates from *S. suffruticosa*.**

| Isolates | Inhibitory zone diameter/mm | | | |
|---|---|---|---|---|
| | *Staphyloccocus aureus* | *Escherichia coli* | *Pseudomonas aeruginosa* | *Enterococcus faecalis* |
| *Chaetomium globosum* | 15.1±0.3 | 21.3±0.5 | 22.4±0.2 | 20.1±0.3 |
| *Chaetomium* sp. 2 | - | 16.3±0.3 | - | - |
| *Chaetomium murorum* | - | - | 3.2±0.3 | 2.1±0.3 |
| *Fusarium* sp. 3 | 18.9±0.8 | 23.4±0.3 | 12.0±0.4 | 13.1±0.5 |
| *Fusarium* sp. 1 | - | 17.3±0.3 | - | - |
| *Fusarium* sp. 2 | 5.8±0.3 | 26.5±0.6 | 14.6±0.3 | 15.0±0.7 |
| *Phomopsis* sp. | - | 17.8±0.7 | - | - |
| *Cladosporium ramotenellum* | 12.1±0.7 | 17.5±0.3 | 12.6±0.7 | 16.3±0.5 |
| *Curvularia* sp. | - | - | 3.2±0.4 | - |
| *Leptosphaeria* sp. | - | 16.4±0.5 | - | - |
| *Penicillium* sp. | - | 13.5±0.3 | - | 12.4±0.3 |

## Determination of alkaloid content

Securinine alkaloids are the main active components of *S. suffruticosa*. Nine species of endophytic fungi contained alkaloids (Table 7). The mass fractions of the alkaloids in *C. globosum* and *Fusarium* sp. 3 reached 0.231% and 0.170%, respectively, thereby indicating high alkaloid content. The endophytic fungal strains *C. globosum* and *Fusarium* sp. 3 showed good antibacterial effects, and the inhibition zones on *E. coli* were larger than 20 mm.

## Discussion

Endophytic fungi have a high degree of taxonomic diversity. These fungi can regulate the morphological and physiological functions of host plants through various mechanisms [36]. These fungi exist in nearly every tissue type studied and are promising as biological control agents against phytopathogens and bioactive substances [37]. Despite these characteristics, endophytic fungi remain poorly studied [38]. Their complex ecological functions and biological resources should be widely developed. *S. suffruticosa*, an important Chinese medicine plant, has attracted increasing attention in the medical and chemical fields because of its active ingredients, but it has received minimal attention in ecological and other research areas.

In our study, 420 endogenous fungi isolated from *S. suffruticosa* roots, stems, and leaves were divided into 56 morphological species, 35 species, 20 genera, 11 orders, four classes, and two phyla. Thus, the endophytic fungi were abundant and showed species richness. This result is similar to that for most plant endophytic fungi. The root, stem, and leaf of *Oryza rufipogon* Griff. yielded 229 endophytic fungi belonging to 19 genera, among which *Pleosporales*, *Phoma*, *Cladosporium*, and *Penicillium* were the dominant ones [39]. A total of 350 strains of endophytic fungi were isolated from the seeds of *Phyllostachys edulis* belonged to 19 genera, in which 98% were ascomycetes and 2% were basidiomycetes [40]. All endophytic fungi are Ascomycetes and Basidiomycetes. Ascomycetes are the most common representatives of endophytic fungal communities isolated using standard separation protocols [41, 42], which supports our finding. Basidiomycetes depend on culture methods [43], and this characteristic explains the low number of Basidiomycota isolates in this work. The culture-dependent method might have overlooked some unculturable endophytic fungi [44], but many isolates could still be separated [39]. We obtained abundant endophytic fungal isolates from *S. suffruticosa*, which was conducive for screening the active strains and laying a foundation for their application.

Most endophytic fungi from the root, stem, and leaf tissues of *S. suffruticosa* belonged to Dothideomycetes and Sordariomycetes. Hypocreales and Sordariales of class Sordariomycetes and Pleosporales of class Dothideomycetes had the highest number of strains and were also rich in species, including the dominant *Chaetomium*, *Fusarium*, and their related genera. Therefore, Sordariales, Pleosporales, and Hypocreales are the most common endophytic fungi in *S. suffruticosa*. *Chaetomium*, *Fusarium*, *Cladosporium*, and *Ceratobasidium*, which are also in other medicinal plants, are the dominant genera of *S. suffruticosa*. The dominant genera in

**Table 6. Determination of the minimum inhibitory concentration (MIC) and minimum bactericidal concentration (MBC) of ethyl-acetate extracts of three endophytic fungal isolates against several pathogenic bacteria.**

| Isolates | MIC/MBC concentration (mg/mL) | | | |
|---|---|---|---|---|
| | *Staphylococcus aureus* | *Escherichia coli* | *Pseudomonas aeruginosa* | *Enterococcus faecalis* |
| *Chaetomium globosum* | 1.00/1.00 | 0.50/0.50 | 0.50/0.50 | 0.50/0.50 |
| *Fusarium* sp. 3 | 1.00/1.00 | 0.50/0.50 | 2.00/2.00 | 2.00/2.00 |
| *Cladosporium ramotenellum* | 2.00/2.00 | 1.00/1.00 | 2.00/2.00 | 1.00/1.00 |

**Table 7. Total alkaloid content of some endophytic fungal isolates from *S. suffruticosa*.**

| Isolates | Total alkaloid content |
|----------|------------------------|
| *Chaetomium globosum* | 0.231% |
| *Chaetomium* sp. 2 | 0.092% |
| *Chaetomium murorum* | 0.100% |
| *Fusarium* sp. 3 | 0.170% |
| *Fusarium* sp. 1 | 0.090% |
| *Phomopsis* sp. | 0.070% |
| *Cladosporium ramotenellum* | 0.129% |
| *Leptosphaeria* sp. | 0.109% |
| *Penicillium* sp. | 0.031% |

the *Rehmannia glutinosa* root system are *Verticillium*, *Fusarium*, and *Ceratobasidium* [45]. The dominant genera of the rhizosphere microorganism include *Fusarium* in the geo- and non-authentic regions of medicinal *Paeonia suffruticosa*, along with *Ceratobasidium* in non-authentic regions. *Ceratobasidium*, *Cladosporium*, and *Fusarium*, which are closely related to the microbial community of medicinal *Echinacea purpurea*, are also present [14]. *Chaetomium* species are also found in many plants, such as *Curcuma wenyujin*, *Platycladus orientalis*, and *Maytenus hookeri* Loes [46,47]. In this work, *S. suffruticosa* was mainly collected from the Shell Islands of Yellow River Delta. Moreover, the soil in this area contain 0.10%–0.21% salt with pH of 8–9 [48]. Moreover, and the composition of *S. suffruticosa* endophytic fungi shows the specificity of the population structure in water-deficient high-salt areas.

The endophytic fungi in *S. suffruticosa* had high diversity. Hence, the researchers analyzed the biodiversity and separately preserved and identified the endophytic fungi in the root, stem, and leaf of *S. suffruticosa*. Significant tissue specificity was confirmed. Four dominant fungi had different specificities, i.e., in the root, stem, and leaf tissues. *Chaetomium*, *Cladosporium*, and *Ceratobasidium* were commonly found in the tissue roots, stems, and leaves. *Chaetomium* was the only dominant endophytic fungus in all these tissues. Fungal strains with strong genetic variation and adaptability can adjust well to changes in the microenvironment and tend to expand their distribution and reproduction in strange places. Species richness and diversity indices reflect a certain degree of heterogeneity due to considerable differences in diversity caused by the microenvironment. The tissue specificity of endophytic fungi in different parts of *S. suffruticosa* may be caused by variations in the microenvironment of plant tissues. Plant characteristics and tissue sampling mainly affect the composition and dynamics of an endophytic fungal community. A study on the microecological distribution of endophytes in different tissues of *Pteroceltis tatarinowii* [49] and *Zanthoxylum bungeanum* [27] produced similar results, which demonstrated the tissue specificity of endophytes.

*F. oxysporum*, *P. herbarum* and *C. siamense* are common pathogens that have been isolated from diseased Euphorbiaceae plants in prior research and from diseased *S. suffruticosa* in our work [50]. Antagonists control pathogens by the production of antimicrobial compounds. Antimicrobial compounds may be produced in volatile and nonvolatile forms [51]. The detection of nonvolatile compounds is one of the simplest and most effective methods of identifying the potential antagonists of pathogenic fungi. We found that endophytic fungi strains *C. globosum*, *Fusarium* sp., and *C. ramotenellum* had enhanced inhibitory effect on pathogens. By comparison with that in the control group, the non-volatile compounds of some strains significantly inhibited the growth of pathogens. The filtrates cultured from these isolates had inhibitory effects on the radial growth of pathogens. This

research verified that nonvolatile antibiotics exist in the filtrates of mycelia. Antibiotic inhibitors are useful in the reducing of plant disease [52].

At present, new antibiotic resources should be identified due to the increasingly serious infection caused by drug-resistant strains. The two main methods currently applied for the development of novel antibiotics are as follows: 1) establishment of a new screening model to screen previously undiscovered active substances in the known resource pool and 2) searching for new repositories. In the past, the screening of microorganisms that produce antimicrobial substances has always emphasized soil microorganisms, especially *Streptomyces*. The ratio of valuable antibiotics screened decreases annually [53].Therefore, researchers have focused on the selection of antimicrobial substances on microorganisms in other ecosystems, particularly those in special habitats, such as marine microorganisms and endophytes [54]. The endophytic fungi of *S. suffruticosa* have become important resources in the search for new drug sources because they grow in the Shell Island of Yellow River Delta and have special habitat and endophytic characteristics. We observed that the extracts from the various endophytic fungi of *S. suffruticosa* had significant inhibitory effects on the growth of multi-drug-resistant pathogenic bacteria *S. aureus*. Moreover, endophytic fungi *C. globosum*, *Fusarium* sp. 3, and *C. ramotenellum* exhibited enhanced bacteriostasis effect, and their endophytic fungi are common in plants. *Chaetomium* is a common plant endophytic fungus that contains various active ingredients and is often used as a biocontrol fungus on plant pathogens. The strain of *Chaetomium* separated from the traditional medicinal plant *Imperata cylindrica* can produce compounds with novel structures and notable pharmacological activities [55]. Chaetoglobosin Vb was separated and identified from the secondary metabolite of endophytic fungus *C. globosum* of *Ginkgo biloba* and has a strong inhibitory effect on various pathogenic bacteria and plant pathogenic fungi that cause severe agricultural damage [56]. The metabolites of endophytic fungus *C. globosum* can inhibit the growth of plant pathogens, and its antioxidant activity can inhibit acetylcholinesterase [57]. *Fusarium* is an endophytic fungus with a high isolation rate and is widely found in plants. A strain of *F. solani* with antimicrobial activity was obtained from the bark of *Taxus chinensis* [58]. *Fusarium* sp. was obtained from the roots of *Mentha longifolia* in Saudi Arabia, and the active metabolites produced can be used for the treatment of fungal infections and malaria [59]. Fusarubin, a metabolite of endophytic fungus *F. solani* from *Glycyrrhiza*, has good inhibitory effects against *Mycobacterium tuberculosis* [60].

In terms of the co-evolution of host plants and endophytes, internal symbiotic theory indicates that in symbiotic organisms, the secondary metabolism in biochemical pathways can be used by other organisms; this condition exhibits interaction and co-evolution [61]. From an evolutionary perspective of fungal evolution, plants and their endophytic fungi are co-evolutionary. Specific endophytic fungi may obtain specific metabolic pathways of host plants due to the transmission of genetic information in long-term co-evolution with host plants, such that endophytic fungi can produce the same or similar physiological active components as host plants. Obtaining natural products with antibacterial activity from plant endophytic fungi can compensate for the shortage of plant resources and the long regeneration cycle and realize the large-scale, low-cost, pollution-free production of natural active compounds through industrialized fermentation. Endophytic fungi are considered new resources of natural antimicrobial compounds, which have high efficiency and are useful for environmental protection [14,62]. Securinega-type alkaloids are the main active components of *S. suffruticosa*. Endophytic fungi *C. globosum* and *Fusarium* sp. 3 have strong antimicrobial activities, and their alkaloid contents reach 0.231% and 0.170%, respectively. Thus, endophytic fungi represent a novel source of new active compounds.

In summary, our findings revealed various endophytic fungi in different tissues of *S. suffruticosa* in Shell Islands of the Yellow River Delta. Fungal endophytes serve not only as biological

control agents that fight the plant pathogens of *S. suffruticosa*, but also as new secondary metabolite resources with biological activities [63]. Of the 35 endophytic fungi, *C. globosum*, *Fusarium* sp. 3 and *C. ramotenellum* exerted significant inhibitory effects on four kinds of pathogenic bacteria and three kinds of plant pathogens and could produce numerous alkaloids. Further experiments that isolate pure compounds and determine their biological activities may generate many new natural compounds from the endophytic fungi derived from *S. suffruticosa*. Results of this study indicated abundant endophytic fungi resources and many potential new species and active strains in the *S. suffruticosa* growing in the Shell Islands of Yellow River Delta. Thus, a foundation for protecting wild *S. suffruticosa* resources and developing new drug leading compounds and biological control agents is established.

## Supporting information

**S1 Fig. PCR amplification results of 18S-ITS1-5.8S-ITS4-28S of some fungal endophytes.**
(TIF)

**S1 Table. The accession number of endophytic fungi.**
(DOC)

**S2 Table. Isolation rate of endophytic fungi from roots, stems and leaves of *S. suffruticosa* species or varieties.**
(DOC)

**S1 File. This is the ITS sequenceof endophytic fungi.**
(PDF)

## Author Contributions

**Conceptualization:** Wen Du, Zhigang Yao, Jialiang Li, Chunlong Sun, Jiangbao Xia.

**Data curation:** Wen Du, Chunlong Sun, Baogui Wang.

**Formal analysis:** Wen Du, Chunlong Sun, Jiangbao Xia.

**Funding acquisition:** Wen Du, Chunlong Sun, Baogui Wang.

**Methodology:** Wen Du, Chunlong Sun.

**Project administration:** Wen Du, Chunlong Sun, Dongli Shi, Lili Ren.

**Resources:** Wen Du, Chunlong Sun.

**Software:** Wen Du, Chunlong Sun.

**Supervision:** Wen Du, Chunlong Sun.

**Validation:** Wen Du, Chunlong Sun.

**Writing – original draft:** Wen Du.

**Writing – review & editing:** Wen Du, Zhigang Yao, Jialiang Li, Chunlong Sun, Jiangbao Xia, Baogui Wang, Dongli Shi, Lili Ren.

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
