## [Decision Letter · Decision Letter 0]

23 Oct 2019

PONE-D-19-22808

Biodiversity and characterization of culturable endophytic fungi isolated from Securinega suffruticosa in Shell Islands of the Yellow River Delta, and analysis of their antimicrobial activity

PLOS ONE

Dear Wen Du and Chunlong Sun,

Thank you for submitting your manuscript to PLOS ONE. After careful consideration, we feel that it has merit but does not fully meet PLOS ONE’s publication criteria as it currently stands. Therefore, we invite you to submit a revised version of the manuscript that addresses the points raised during the review process.

Please, see below comments:

1. Professional English proofreading of your manuscript is necessary before resubmission; 2. Shorten the title of your manuscript; 3. Rewrite Introduction - paragraphs describing endophytes and Securinega suffruticosa - *t**he same beginning of each sentence, e.g. *Endophytic fungi and S. suffruticosa.; 4. Change manuscript according to the reviewers suggestions.

We would appreciate receiving your revised manuscript by Dec 07 2019 11:59PM. To enhance the reproducibility of your results, we recommend that if applicable you deposit your laboratory protocols in protocols.io, where a protocol can be assigned its own identifier (DOI) such that it can be cited independently in the future. For instructions see: http://journals.plos.org/plosone/s/submission-guidelines#loc-laboratory-protocols

We look forward to receiving your revised manuscript.

Kind regards,

Katarzyna Hrynkiewicz

Academic Editor

PLOS ONE

Journal Requirements:

1.

12 This work was financially supported by Shandong Provincial Natural Science

13 Foundation (ZR2016BL16, ZR2016CL01); Doctor Foundation of Binzhou University

14 (2016Y17, 2016Y02); Science and Technology Development Plan Project of

15 Shandong Province (2014GSF119032); Project of Shandong Province Higher

16 Educational Science and Technology Program (J17KA120); Research Project of the

17 University-level teaching Reform of Binzhou University in 2017 (BYJYYB201736);

18 School-enterprise Co-construction Course Project of Binzhou University in 2017

19 (BYXQGJ201706), China.

Please remove any funding-related text from the manuscript and let us know how you would like to update your Funding Statement. Currently, your Funding Statement reads as follows: "Yes"

Reviewers' comments:

Reviewer's Responses to Questions

**Comments to the Author**

1. Is the manuscript technically sound, and do the data support the conclusions?

Reviewer #1: No

Reviewer #2: Partly

2. Has the statistical analysis been performed appropriately and rigorously? 

Reviewer #1: No

Reviewer #2: Yes

3. Have the authors made all data underlying the findings in their manuscript fully available?

Reviewer #1: No

Reviewer #2: Yes

4. Is the manuscript presented in an intelligible fashion and written in standard English?

Reviewer #1: Yes

Reviewer #2: No

5. Review Comments to the Author

Reviewer #1: Methodology:

a) The samples of Securinega suffruticosa were collected during two sampling years (2017 and 2018). It would be interesting to show the results of the culturable endophytic fungal composition between the two sampling periods.

b) Secondly the protocol for endophyte isolation used in this study does not guarantee that all the culturable fungal diversity was obtained. I suggest that in addition to plating the 10 cm tissue sections on PDA agar, you could also crush the tissue samples using motor and pestle and perform serial dilutions. This will allow more surface area for the fungi to grow (slow growing fungi can be obtained efficiently) which may have not been isolated as the fast growing fungi dominates the incision/cut area of the 10 cm tissue.

You can refer to the summary of protocols for endophyte isolation given by “Verma S.K., Kharwar R.N., Gond S.K., Kingsley K.L., White J.F. (2019) Exploring Endophytic Communities of Plants: Methods for Assessing Diversity, Effects on Host Development and Potential Biotechnological Applications. In: Verma S., White, Jr J. (eds) Seed Endophytes. Springer, Cham.”

Results:

a) As stated above, your results could also show the number of fungal isolates obtained between sampling periods.

b) Statistical inferences (e.g. ANOVA) needs to be generated for the data on the dominance (Y) values of endophytic fungi in tissues, antagonism of endophytic fungal isolates, etc.

Reviewer #2: In the manuscript presented for review, the authors present the results of experiments concerning the isolation and partial characterization of endophytic fungi from Securinega suffruticosa. The initial number of isolates was reduced to 57 for which ITS region sequences were obtained.

The description of the research makes it possible to repeat it, but I have a few questions about the methodology.

P7; L4-5 - Whether the strains came from public accessible collections of microorganisms and had identification numbers? Where they a type strains? What was the source of Fusarium oxysporum and Phoma herbarum?

P8; L6-8 – What was the result of surface disinfection? Is it possible to avoid the fungal growth on PDA medium when the surface sterilised parts of plant contain endophytic fungi and their release from tissues is very probable? Why the water from the last rinse was not tested for fungal presence? How long was the control incubated?

P11; L17-18 - What method was used to determine the number of bacteria in the inoculum?

P12; L20 – Is it possible to obtain more specified than total alkaloid content data? Chromatographic data (e.g. securinice) are strongly recommended.

Results:

P15; L7-8 vs. P14; L3-4 – In my opinion the results obtained from phylogenetic studies concern 57 isolates (57 ITS sequences were obtained, not 420). Morphological analysis does not allow for unambiguous classification of isolates.

P18; L19-22 and P19; L1-8 - I suggest rebuilding the entire chapter because it is currently written in a chaotic manner

P19; L1 – Table 5

P19; L5 – the description does not correspond with data presented in Table 6, e.g. “the crude extracts of C. globosum and C. ramotenellum (1.00 mg/mL) were enough to inhibit S. aureus”. For C. ramotenellum the MIC/MBC values are 2.00/2.00.

Strengths of the article:

A large number of isolates.

Diversity analyzes of endophytic fungi.

Interesting discussion of the results .

Weaker manuscript traits:

Only total alkaloid content of some endophytic fungal isolates.

Methodological doubts that require clarification.

The quality of the text should be assessed by native speaker. I am not, but I see fragments that are difficult to understand (especially in the description of the results).

6. PLOS authors have the option to publish the peer review history of their article (what does this mean?). If published, this will include your full peer review and any attached files.

Reviewer #1: No

Reviewer #2: No

---

## [Author Response · Author response to Decision Letter 0]

30 Dec 2019

Dear Editor and Reviewers:

All authors agree to submit the work to PLOS ONE; the work has not been published and submitted to another journal. The order of author is Wen Du, Zhigang Yao, Jialiang Li, Chunlong Sun, Jiangbao Xi, Baogui Wan, Dongli Shi, Lili Ren. Wen Du is first author; Wen Du and Chunlong Sun are corresponding authors. Endophytic fungi in Securinega suffruticosa is the focus of research, so many researchers attempted to study it. I hope to expedite the review of my manuscript. 

Thank you for your letter and for the reviewers’ comments concerning our manuscript entitled “Diversity and antimicrobial activity of endophytic fungi isolated from Securinega suffruticosa in the Yellow River Delta” (Manuscript Number: PONE-D-19-22808). Those comments are all valuable and very helpful for revising and improving our paper, as well as the important guiding significance to our researches. We have studied comments carefully and have made correction which we hope to meet with approval. The original article has been basically changed, and the modification is too large. 

The following are responses to the reviewer's comments on revision.

Responds to the editor’s comments:`

1.Professional English proofreading of your manuscript is necessary before resubmission; Rewrite Introduction - paragraphs describing endophytes and Securinega suffruticosa - the same beginning of each sentence, e.g. Endophytic fungi and S. suffruticosa.;

Response: We rewrote the preface. The English translation company has revised it.

2.Shorten the title of your manuscript; 

Response: The title changed from “Biodiversity and characterization of culturable endophytic fungi isolated from Securinega suffruticosa in Shell Islands of the Yellow River Delta, and analysis of their antimicrobial activity” to “Diversity and characterization of endophytic fungi isolated from Securinega suffruticosa in the Yellow River Delta, and analysis of their antimicrobial activity”. 

Reviewer #1: Methodology:

a) The samples of Securinega suffruticosa were collected during two sampling years (2017 and 2018). It would be interesting to show the results of the culturable endophytic fungal composition between the two sampling periods.

Response: The another supplementary experiment was conducted to make the whole experiment more complete in 2018.

b) Secondly the protocol for endophyte isolation used in this study does not guarantee that all the culturable fungal diversity was obtained. I suggest that in addition to plating the 10 cm tissue sections on PDA agar, you could also crush the tissue samples using motor and pestle and perform serial dilutions. This will allow more surface area for the fungi to grow (slow growing fungi can be obtained efficiently) which may have not been isolated as the fast growing fungi dominates the incision/cut area of the 10 cm tissue.

Response: It has been cut into small sections of 10 cm before disinfection, and then cut into small pieces of 0.5cm × 0.5cm after disinfection. It is hoped that more endophytic fungi can still be isolated after removing the microorganisms on the surface.

Results:

a) As stated above, your results could also show the number of fungal isolates obtained between sampling periods.

Response: There was at first, but it was limited by the length of the article. Moreover, this part of data is only the increase of workload and lacks certain depth.

b) Statistical inferences (e.g. ANOVA) needs to be generated for the data on the dominance (Y) values of endophytic fungi in tissues, antagonism of endophytic fungal isolates, etc.

Response: When statistical calculation is needed, we try to add as much as possible, but it is not reasonable to add anova on the dominant value and antagonism.

Reviewer #2: In the manuscript presented for review, the authors present the results of experiments concerning the isolation and partial characterization of endophytic fungi from Securinega suffruticosa. The initial number of isolates was reduced to 57 for which ITS region sequences were obtained.

The description of the research makes it possible to repeat it, but I have a few questions about the methodology.

P7; L4-5 - Whether the strains came from public accessible collections of microorganisms and had identification numbers? Where they a type strains? What was the source of Fusarium oxysporum and Phoma herbarum?

Response: The test strains were provided by the Biopharmaceutical Center of Binzhou University and the institute of Biochemistry and Nutrition of Guizhou University. The test strains included Fusarium oxysporum and Phoma herbarum.

P8; L6-8 – What was the result of surface disinfection? Is it possible to avoid the fungal growth on PDA medium when the surface sterilised parts of plant contain endophytic fungi and their release from tissues is very probable? Why the water from the last rinse was not tested for fungal presence? How long was the control incubated?

Response: We hope to remove the microorganisms on the surface of the plant as much as possible, and the control was cultured for 7 days. If some endophytic fungi released by plant tissues are sterilized and removed, the endophytic fungi can also be present in plant tissues.

P11; L17-18 - What method was used to determine the number of bacteria in the inoculum?

Response: Take a small amount to measure the concentration through microscope, add sterile water to adjust the concentration.。

P12; L20 – Is it possible to obtain more specified than total alkaloid content data? Chromatographic data (e.g. securinice) are strongly recommended.

Response: We will study the metabolites of individual strain. This experiment is mainly about “Biodiversity and characterization of culturable endophytic fungi and analysis of their antimicrobial activity”. Research papers also have a theme.

Results:

P15; L7-8 vs. P14; L3-4 – In my opinion the results obtained from phylogenetic studies concern 57 isolates (57 ITS sequences were obtained, not 420). Morphological analysis does not allow for unambiguous classification of isolates.

Response: Morphological analysis does not provide a clear classification. So we did the molecular identification.

P18; L19-22 and P19; L1-8 - I suggest rebuilding the entire chapter because it is currently written in a chaotic manner

P19; L1 – Table 5

P19; L5 – the description does not correspond with data presented in Table 6, e.g. “the crude extracts of C. globosum and C. ramotenellum (1.00 mg/mL) were enough to inhibit S. aureus”. For C. ramotenellum the MIC/MBC values are 2.00/2.00.

Response: We reorganized the chapter, and there were some problems with irregularities.

We tried our best to improve the manuscript and made some changes in the manuscript. We appreciate for Editors/Reviewers’ warm work earnestly, and hope that the correction will meet with approval.

Once again, thank you very much for your comments and suggestions.

---

## [Editor Report · Decision Letter 1]

11 Feb 2020

Diversity and antimicrobial activity of endophytic fungi isolated from Securinega suffruticosa in the Yellow River Delta

PONE-D-19-22808R1

Dear Dr. Sun,

We are pleased to inform you that your manuscript has been judged scientifically suitable for publication and will be formally accepted for publication once it complies with all outstanding technical requirements.

With kind regards,

Katarzyna Hrynkiewicz

Academic Editor

PLOS ONE
---

## [Editor Report · Acceptance letter]

25 Feb 2020

PONE-D-19-22808R1 

Diversity and antimicrobial activity of endophytic fungi isolated from *Securinega suffruticosa* in the Yellow River Delta 

Dear Dr. Sun:

I am pleased to inform you that your manuscript has been deemed suitable for publication in PLOS ONE. Congratulations! Your manuscript is now with our production department. 

With kind regards,

on behalf of

Dr. Katarzyna Hrynkiewicz 

Academic Editor

PLOS ONE